# CAMPYAIR, a New Selective, Differential Medium for *Campylobacter* spp. Isolation without the Need for Microaerobic Atmosphere

**DOI:** 10.3390/microorganisms10071403

**Published:** 2022-07-12

**Authors:** Arturo Levican, Arthur Hinton

**Affiliations:** 1Tecnología Médica, Pontificia Universidad Católica de Valparaíso, Avenida Universidad 330, Valparaíso 2373223, Chile; 2Poultry Microbiological Safety and Processing Research Unit, U.S. National Poultry Research Center, Agricultural Research Service, Athens, GA 30605, USA; arthur.hinton@usda.gov

**Keywords:** *Campylobacter*, aerobic incubation, solid medium, selective, differential, CAMPYAIR

## Abstract

*Campylobacter* spp. are considered the most frequent bacterial cause of acute gastroenteritis worldwide. Although the diarrhea produced by these bacteria is self-limiting, the pathogen has been associated with severe long-term sequelae following acute signs and symptoms of the illness. However, research on *Campylobacter* is hampered by costs and technical requirements for isolating and culturing the bacterium, especially in low and middle-income countries. Therefore, attempts have been made to simplify these culture methods and to reduce costs associated with conducting research on *Campylobacter*. Recently, a liquid medium which allows selective enrichment of *Campylobacter* using aerobic incubation has been described. However, a solid medium is also needed for the isolation of pure colonies, enumeration of bacterial populations, and other studies on the pathogen. Therefore, a new medium (CAMPYAIR) was developed, based on the formulation of the liquid medium. CAMPYAIR is a solid chromogenic medium that supports the growth of *Campylobacter* isolates within 48 h of incubation in aerobic atmospheres. Moreover, CAMPYAIR contains antibiotic supplements with an enhanced ability to recover *Campylobacter* from environmental samples that may also contain non-campylobacter bacteria. The addition of the indicator 2,3,5-triphenyltetrazolium (TTC) to the medium differentiates *Campylobacter* from other bacteria growing on the media. The findings from studies on CAMPYAIR suggest that the utilization of the new selective, differential medium could help to reduce the costs, equipment, and technical training required for *Campylobacter* isolation from clinical and environmental samples.

## 1. Introduction

*Campylobacter* spp., in particular thermotolerant species such as *Campylobacter jejuni* and *Campylobacter coli*, have been identified as major concerns in food safety and general public health. In fact, *Campylobacter* is responsible for more human food-borne illnesses than any other bacterium worldwide [1,2,3,4]. It is estimated that *C. jejuni* and *C. coli* produced between 400 and 500 million cases of campylobacteriosis per year in the world [5]. Risk factors for *Campylobacter* infection are mostly associated with foodborne transmission routes and poor food hygiene and handling, particularly with the consumption of undercooked poultry. However, during the COVID pandemic, there was a decrease in laboratory reporting of *Campylobacter* and other gastrointestinal pathogens. This decrease may have been due to the prioritization of testing for SARS-CoV-2 testing and a decrease in testing for causes of foodborne illnesses [6].

Although the diarrhoea produced by *Campylobacter* and other enteropathogens are usually self-limiting and do not require specific treatment, some complications have been associated with campylobacteriosis such as septicaemia, meningitis, haemolytic uremic syndrome, pancreatitis, and abortions. Additionally, reactive arthritis and syndromes of Guillain-Barré (GBS) have been identified as sequela of the illness [2,7]. Furthermore, Toledano et al. [8]) reported three cases of severe infections caused by *Campylobacter* spp. in which symptoms mimicked a Multisystem Inflammatory Syndrome in children (MIS-C), normally associated with COVID-19. On the other hand, increasing rates of resistance to fluoroquinolones, tetracycline, and even erythromycin which is the treatment of choice have been observed; therefore, the WHO does not recommend the use of empirical therapies for *Campylobacter* spp. [8]. In this line, this organization also recommends the use of culturing over culture-independent diagnostic tests (CIDTs) because it provides isolates for subtyping and resistance monitoring that can produce useful information for further public health action [9].

Research on *Campylobacter* is hampered by the costs and technical requirements associated with the isolation and culture of *Campylobacter* spp., especially in low- and middle-income countries [10]. Therefore, attempts have been made to simplify culture methods by the formulation of blood-free media and other media that allow the growth of the pathogen without the need for producing artificial, microaerobic atmospheres [11]. Some research has shown that it is possible to grow the bacterium under aerobic conditions in a liquid medium supplemented with a mixture of organic acids associated with the Krebs cycle, vitamins, minerals, and sodium bicarbonate [12]. Later research indicated that liquid medium was improved by adding soy extracts and peptones [9]. Based on those studies, Hinton et al. [13] proposed a selective liquid medium intended to be used in the recovery of *Campylobacter* spp. Under aerobic conditions. The improved medium included high concentrations of organic and amino acids, as well as selective antibiotics supplements that had been previously used in media such as Cefex, Bolton, and Skirrow [13].

The liquid medium described above may be used for selective enrichment of *Campylobacter* spp. from food and environmental samples; however, isolation of colonies and other microbiological techniques can only be performed by using a solid, agar medium dispensed in Petri dishes. In the present study, the selective liquid medium was modified to formulate a novel solid medium base which allows for obtaining growth of *Campylobacter* spp. under aerobic conditions. For this objective, the recovery of *C. jejuni*, *C. coli,* and other *Campylobacter* isolates recovered from clinical samples on CAMPYAIR was determined. Additionally, different antibiotic supplements were tested for the ability to increase the selectivity of the new medium by inhibiting the growth of non-*Campylobacter* while enhancing the recovery of *Campylobacter* spp. from complex matrices such as human faeces or foods.

## 2. Materials and Methods

### 2.1. Preparation of Non-Selective Basal Solid Medium

The non-selective basal medium was based on Hinton and Cox’s [13] medium, which was composed of (g/L) beef extract (Merck Millipore, Burlington, MA, USA; Catalog Number (CN) B4888); 50.0 g/L, tryptose or tryptone (Sigma-Aldrich Co., St. Louis, MO, USA; CN 70937); 10.0 g/L, sodium lactate, syrup, 60% *w*/*v* (Sigma-Aldrich Co., USA; CN L1375); 3.0 mL/L, sodium bicarbonate 1.5 g/L (Merck Millipore, Burlington, MA, USA; CN 1.06329); and 900 mL of distilled water. The medium was prepared as described by the authors. The solid medium was prepared by adding concentrations of 1.0%, 1.5%, and 2.0% of agar-agar (Liophilchem, Roseto degli Abruzzi, TE, Italy; CN 611001) to the medium. Additionally, the growth of *Campylobacter* on solid media containing 0.25%, 0.5%, 1.0%, and 2.0% concentrations of soluble starch (Sigma-Aldrich Co., St. Louis, MO, USA; CN S9765) was also determined.

Other supplements were also examined for the ability to improve the growth or detection of *Campylobacter* on the medium. The medium was supplemented with sodium deoxycholate (Merck Millipore, Burlington, MA, USA; CN 106504) at 0.025%, 0.05%, 0.1%, and 0.2% and supplementation with 10% defibrinated sheep blood was also tested. In addition, the indicator 2,3,5-triphenyltetrazolium (TTC, Sigma-Aldrich Co., St. Louis, MO, USA; CN T8877) was used at a concentration of 200 μg/mL to facilitate recognition of *Campylobacter* colonies [14].

The basal medium was prepared by dissolving all the components, except sodium bicarbonate, in distilled water. The medium was sterilized by autoclaving at 121 °C for 15 min. The autoclave media was tempered to 55 °C and 900 mL aliquots of medium were supplemented with 100 mL of 1.5% sodium bicarbonate that had been filter sterilized by passing through a 0.2 µm filter. The basal medium that was supplemented with blood and sodium bicarbonate was prepared in 800 mL aliquots and supplemented with 100 mL of 1.5% sodium bicarbonate and 100 mL of defibrinated sheep blood.

The supplemented medium was distributed in sterile Petri dishes and allowed to cool. Plates with solidified media were sealed with parafilm or plastic tape (3M, Saint Paul, MN, USA) to avoid the interchange of oxygen with the environment. Furthermore, and very importantly, after inoculating the surface of the solid medium with the *Campylobacter* isolates, Parafilm^®^ or plastic tape was reapplied to the plates to seal them.

### 2.2. Selective Media Supplements

The effect of supplementing the media with selective antibiotics on the growth of bacteria on the media was examined by comparing bacterial growth on the media with the new medium not supplemented with antibiotics to the growth of the bacteria on media that had been supplemented with antibiotics after aerobic incubation at 37 C for 48 h. Modifications of two antibiotic mixtures used as selective *Campylobacter* media supplements were tested. One selective supplement tested was the CCDA supplement that contained 32 mg/L of cefoperazone and 10 mg/L of amphotericin B (Lyophilchem, Roseto degli Abruzzi, TE, Italy; CN 81037), with and without the addition of 10 mg/L or 20 mg/L of vancomycin hydrochloride (Sigma-Aldrich Co., St. Louis, MO, USA; CNs 75423). The other supplement was the Skirrow antibiotic supplement that contained 10 mg/L of vancomycin hydrochloride, 20 mg/L of trimethoprim lactate, and 2500 IU polymyxin B 2500 IU (Sigma-Aldrich Co., St. Louis, MO, USA; CNs T0667, and 1405-20-5, respectively). Concentrations listed for each supplement are the final concentration of antibiotics after addition to the medium.

### 2.3. Campylobacter Isolates

Growth of *Campylobacter* on the solid medium under aerobic incubation, with and without the antibiotics supplements listed above was tested. The growth of *C. jejuni*, *C. coli*, and ten *Campylobacter* field strains obtained from clinical samples [15] on the media were examined. All bacterial isolates were streaked onto blood agar (Trypticase soy agar, Oxoid, Basingstoke, UK) supplemented with 5% defibrinated sheep blood and incubated under microaerobic conditions at 42 °C for 48 h.

Non-*Campylobacter* bacteria used to determine the selectivity of the medium were *Staphylococcus aureus*, *E. coli* ATCC, and *Pseudomonas aeruginosa*.

Bacterial suspension of approximately 10^8^ CFU/mL was prepared by suspending bacterial colonies of the cultures in 0.85 NaCl to achieve an optical density of 0.05 at 520 nm. For qualitative analyses, a 10 µL aliquot of the bacterial suspensions was streaked directly onto the surface of agar media using a calibrated loop to perform quadrant streaks. After incubation, growth observed in only one quadrant (1+) was defined as “low”; growth observed in two quadrants (2+) was defined as “regular”; while bacterial growth observed in three or four quadrants (3+) was considered “abundant”. In addition, the selectivity of the medium was also qualitatively assessed by culturing a prepared sample of human faeces artificially inoculated with *Campylobacter* and non-*Campylobacter* isolates. For this, 4 g of non-diarrheic faeces were inoculated with 100 µL of each bacterial suspension of *C. jejuni*, *Staphylococcus aureus*, *E. coli*, and *Pseudomonas aeruginosa* prepared as described above. The sample was homogenized and then used to inoculate triplicate plates of the new medium and incubated aerobically for 48 h at 42 °C

Bacteria were enumerated using the Miles and Misra method [16] as described by Levican et al. [17].

All the experiments were performed in triplicate. BA plates were streaked as controls to demonstrate the growth of *Campylobacter* and other non-*Campylobacter* isolates incubated microaerobically at 42 °C for 48 h.

## 3. Results

Growth of the strain *C. jejuni* on medium supplemented with 1.0%, 1.5%, and 2% agar-agar was tested, but there was no difference in *Campylobacter* growth on media containing either agar-agar concentration. Therefore, all remaining studies were conducted using media containing 1.5% agar-agar (Table 1). Results of studies that examined the bacterial growth on media containing different concentrations of soluble starch indicated that the best *Campylobacter* growth was obtained on media with starch concentrations of 0.5% and 1.0% (Table 1); therefore, the 1.0% concentration was selected for further analyses.

Results of the effect of different concentrations of sodium deoxycholate on the growth of *Campylobacter* on the media incubated aerobically are in Table 1. Findings indicated that media supplemented with 0.1% deoxycholate produced colonies with regular and abundant *Campylobacter* growth. Then, the basal medium was supplemented with 1.5% agar-agar, 1.0% soluble starch, and 0.1% deoxycholate. Furthermore, 10% of defibrinated sheep blood was also added to enhance the growth of the *Campylobacter* strains. The newly formulated medium has been named “CAMPYAIR” and was subjected to further testing.

Moreover, the addition of TTC to CAMPYAIR media allows the recognition of *Campylobacter* spp. colonies which produce a purple colour with or without metallic sheen when growing on the medium even though blood agar is red (Figure 1). Petri plates were initially sealed with Parafilm^®^, however, a better performance was observed when plates were sealed with plastic adhesive tape because of the reduced interchange of gases allowed by the latter. Additionally, although results were recorded after 48 h incubation, all plates were also observed after 24 h incubation, and by that time, only slight growth was observed (data not shown).

To demonstrate the capacity of the new medium, CAMPYAIR, to support the aerobic growth of the *Campylobacter* isolates, serial dilutions of the strains *C. jejuni* and *C. coli* were carried out in parallel to BA incubated under microaerobic conditions. The bacterial counts obtained using the Miles and Misra method [16] showed similar performance for both media (Table 2).

The qualitative growth of *Campylobacter* isolates obtained from clinical samples in a previous study [15] were assessed in the CAMPYAIR medium incubated under aerobic conditions, compared with BA incubated under microaerobic conditions (Table 3) and CAMPYAIR showed a similar performance to BA.

Finally, the inhibition of other bacteria by antibiotic supplements was tested. The best inhibition was obtained with the CCDA supplement plus 10 μg/L vancomycin compared to CCDA alone or the Skirrow supplement (Table 4). The selectivity of the new medium was also qualitatively assessed by culturing a prepared sample of human faeces inoculated with *C. jejuni*, *S. aureus*, *E. coli*, and *P. aeruginosa*. The *C. jejuni* strain showed growth on CAMPYAIR (1+,1+, 1+) and no growth of other bacteria present naturally or artificially in the faecal sample was observed.

## 4. Discussion

The liquid medium previously described by Hinton and Cox [13] was modified in this study to produce a solid medium by adding 1.5% of agar-agar, 1.0% of soluble starch, 0.1% of deoxycholate, 10 μg/L TTC, and 10% of sheep blood. The new medium was named “CAMPYAIR” and it supported the aerobic *Campylobacter* growth after 48 h of incubation. Field strains of *C. jejuni* and *C. coli*, which are the species most associated with foodborne cases of human gastroenteritis, were able to grow on this medium with aerobic incubation. Furthermore, the addition of the indicator TTC allowed easy recognition of *Campylobacter* colonies growing on the medium. Furthermore, the addition of the antibiotic supplement CCDA and 10 μg/L vancomycin increased its selectivity and allowed *Campylobacter* to be isolated from human faecal samples In this line, the isolation of *Campylobacter* from a prepared faecal sample demonstrated the ability of the medium to isolate *Campylobacter* from complex samples using this antibiotics supplement. Altogether, these findings suggest that the utilization of CAMPYAIR could help to reduce the costs, equipment, and technical training required for *Campylobacter* isolation from clinical or environmental samples. The new medium could increase the number of laboratories able to conduct research on *Campylobacter* and especially in low-income countries. However, the capacity of CAMPYAIR to recover field isolates of *Campylobacter* spp. from complex matrices such as human faces or foods should be assessed in future studies in comparison with other selective mediums commonly used. Moreover, the ability of this medium to support the growth of other *Campylobacter* spp. other than *C. jejuni* and *C. coli* should also be examined.

CAMPYAIR may support the growth of *Campylobacter* in primary containers without microaerobic incubation because of the concentration of CO_2_ released by the medium and by additional CO_2_ released by *Campylobacter* growing in the medium (Hinton, personal communication). Sealing of the plate prevents the exchange between the atmosphere inside of the plate and the aerobic atmosphere outside of the plate. Therefore, this also helps to decrease the concentration of oxygen in the primary container. The resistance to oxygen could also be related to the production of pyruvate in the medium. Pyruvate is widely recognized as a scavenger of reactive oxygen species (ROS) because it attenuates the H_2_O_2_-induced ROS formation in the culture medium [18]. Moreover, pyruvate embodies antioxidant properties due to its α-ketocarboxylate structure, enabling it to directly neutralize peroxides and peroxynitrite. The metabolism of pyruvate also generates NADPH that can maintain glutathione-reducing power [19]. Even though CAMPYAIR does not contain pyruvate in its formulation, this organic acid has been observed to have a central role in the metabolism of the citric-acid cycle in *Campylobacter*. Pyruvate may be produced by the metabolism of serine, lactate, and L-fucose [20], therefore, CAMPYAIR organic acid may be produced by of metabolism of lactate and amino acids present in the formula. Moreover, previous studies have demonstrated that *Campylobacter* has some oxygen-regulated genes that increase their expression under aerobic conditions. These genes include several citric-acid cycle enzymes, that become more abundant with increasing aerobiosis [21]. In this line, we observed only slight growth after 24 h of incubation and a better growth after 48 h. This is probably because bacterial cells grow slowly under aerobic conditions until their citric-acid cycle is upregulated producing a higher intracellular concentration of pyruvate for ROS scavenging. Additional studies will be conducted to improve the ability of CAMPYAIR to isolate *Campylobacter* from environmental samples and to improve the utilization of the media for food safety and medical studies.

## Figures and Tables

**Figure 1 microorganisms-10-01403-f001:**
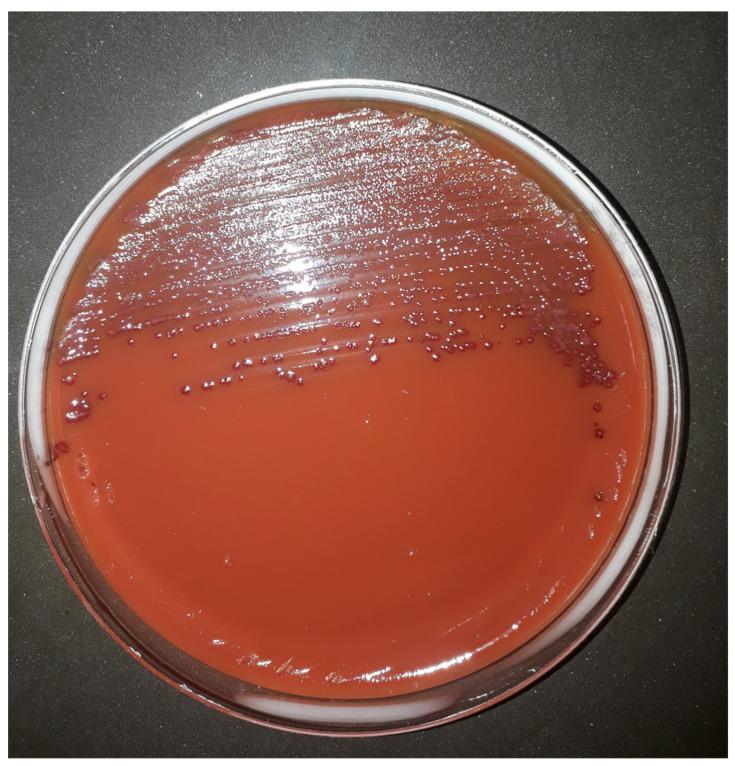
Strain *C. jejuni* (ATCC 33560^T^) was streaked on CAMPYAIR and incubated at 42 °C under aerobic conditions for 48 h.

**Table 1 microorganisms-10-01403-t001:** Qualitative growth of *C. jejuni* ATCC 33560^T^ incubated under aerobic or microaerobic conditions.

	Concentration of agar-agar				
	2.0%	1.5%	1.0%	BA (control)	
Aerobic	2+/2+/1+	**2+/2+/2+**	2+/2+/2+	NG	
Microaerobic	3+/3+/3+	3+/3+/3+	3+/3+/3+	3+/3+/3+	
	Concentration of starch				
	2.0%	1.0%	0.5%	0.25%	BA (control)
Aerobic	1+/NG/1+	**2+/2+/1+**	2+/2+/1+	NG/1+/1+	NG
Microaerobic	**3+/3+/3+**	**3+/3+/3+**	**3+/3+/3+**	**3+/3+/3+**	3+/3+/3+
	Concentration of Sodium deoxycholate				
	0.2%	0.1%	0.05%	0.025	BA (control)
Aerobic	1+/NG/1+	**3+/2+/3+**	2+/2+/1+	2+/1+/1+	NG
Microaerobic	3+/3+/3+	3+/3+/3+	3+/3+/3+	3+/3+/3+	3+/3+/3+

1+, low growth; 2+, regular growth; 3+, abundant growth; NG, no growth; BA, blood agar (control medium). The concentration of each component selected is in bold.

**Table 2 microorganisms-10-01403-t002:** Quantitative growth (CFU/mL) of strains *C. jejuni* (ATCC 33560^T^) and *C. coli* (DSM 4689^T^) on BA incubated under microaerobic conditions and on CAMPYAIR incubated under aerobic conditions. Bacteria were enumerated using the Miles and Misra method.

Strain	Species	BA/Microaerobic	CAMPYAIR/Aerobic
ATCC 33560^T^	*C. jejuni*	3.5 × 10^8^ 9.0 × 10^8^ 1.0 × 10^8^	2.0 × 10^9^ 3.0 × 10^8^ 3.0 × 10^8^
DSM 4689^T^	*C. coli*	1.8 × 10^8^ 4.2 × 10^8^ 1.2 × 10^8^	1.2 × 10^9^ 3.1 × 10^8^ 2.2 × 10^8^

**Table 3 microorganisms-10-01403-t003:** Qualitative growth of strains *C. jejuni* (ATCC 33560^T^) and *C. coli* (DSM 4689^T^) and 10 clinical strains on BA incubated under microaerobic conditions and on CAMPYAIR incubated under aerobic conditions.

Strain	Species	CAMPYAIR/Aerobic	BA/Microaerobic
DSM 4688^T^	*C. jejuni*	3+/3+/3+	3+/3+/3+
DSM 4689^T^	*C. coli*	3+/2+/3+	3+/3+/3+
TM-PUCV 1	*C. jejuni*	3+/3+/3+	3+/3+/3+
TM-PUCV 2	*C. jejuni*	3+/3+/3+	3+/3+/3+
TM-PUCV 3	*C. jejuni*	3+/3+/3+	3+/3+/3+
TM-PUCV 4	*C. jejuni*	3+/3+/3+	3+/3+/3+
TM-PUCV 5	*C. jejuni*	3+/3+/3+	3+/3+/3+
TM-PUCV 6	*C. jejuni*	3+/3+/3+	3+/2+/3+
TM-PUCV 7	*C. coli*	3+/3+/3+	3+/3+/3+
TM-PUCV 8	*C. coli*	3+/3+/3+	3+/3+/3+
TM-PUCV 9	*C. jejuni*	3+/3+/3+	3+/3+/3+
TM-PUCV 10	*C. jejuni*	3+/2+/3+	3+/3+/3+

1+, low; 2+, regular; 3+, abundant growth; BA, blood agar.

**Table 4 microorganisms-10-01403-t004:** Qualitative growth of strains *C. jejuni* (ATCC 33560^T^), *C. coli* (DSM 4689^T^), *S. aureus* ATCC 25923, *E. coli* ATCC 25922, and *P. aeruginosa* ATCC 27853 on CAMPYAIR incubated under aerobic conditions and supplemented with different antibiotic formulations.

	Antibiotic Formulation				
	CCDA	CCDA + 10 μg/L Van	CCDA + 20 μg/L Van	Skirrow	BA/Aerobic
DSM 4688^T^	3+/3+/3+	3+/3+/3+	3+/3+/3+		NG
DSM 4689^T^	3+/3+/3+	3+/3+/3+	3+/2+/3+		NG
TM-PUCV 1	3+/3+/3+	3+/2+/3+	3+/2+/3+		NG
*E. coli*	NG	NG	NG	NG	3+/3+/3+
*P. aeruginosa*	NG/1+/NG	NG	NG	NG/1+/NG	3+/2+/3+
*S. aureus*	1+/1+/NG	NG	NG	NG	3+/3+/3+

Van, vancomycin; 1+, low; 2+, regular; 3+, abundant growth; NG, no growth; BA, blood agar (control).

## Data Availability

Not applicable.

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
