# Peer review of "CAMPYAIR, a New Selective, Differential Medium for Campylobacter spp. Isolation without the Need for Microaerobic Atmosphere"

_microorganisms, 2022, doi:10.3390/microorganisms10071403_

Round 1
Reviewer 1 Report
The development of methods for easier isolation of Campylobacter spp. is crucial in estimating foodborne cases caused by this pathogen.
The study was well described and planned. However, some minor corrections should be made:
lines 30 and 31 - I suggest changing "produce" to "are implicated in"
ex. lines 40, 69, 72, 88, 131, 162 - please write in italics all names of the bacterial genus
lines 120, 201, 207 - please use upper or lower letters for numbers
line 121 - I understand that it was optical density equivalent to 0.5 McFarland standard
line 141 - the end of this sentence is missing (the effect of ......?)
line 105 - the name of one antibiotic was written in capital letter and the second in small letter
line 82 - double "of"
line 86 - text shift
Author Response
The development of methods for easier isolation of Campylobacter spp. is crucial in estimating foodborne cases caused by this pathogen.
The study was well described and planned. However, some minor corrections should be made:
Answer: Thank you for your comments, your suggestions have been assessed as follows (yellow in the new version) :
lines 30 and 31 - I suggest changing "produce" to "are implicated in"
- lines 40, 69, 72, 88, 131, 162 -please write in italics all names of the bacterial genus
Answer: The suggested changes were done.
lines 120, 201, 207 - please use upper or lower letters for numbers
Answer: The appropriate superscripts and subscripts have been added to the text.
line 121 - I understand that it was optical density equivalent to 0.5 McFarland standard
Answer: It was a mistake, it is not 0.5 but 0.05 OD at 520 nm and this was fixed in the new version. Thank you for notice this.
line 141 - the end of this sentence is missing (the effect of ......?)
Answer: The sentence has been completed.
line 105 - the name of one antibiotic was written in capital letter and the second in small letter
Answer: This has been changed.
line 82 - double "of"
line 86 - text shift
Answer: Both changes have been done.
Reviewer 2 Report
The Authors proposed interesting research on the use of a new medium for Campylobacter spp. Isolation without the need for microaerobic atmosphere. Despite the paper is adequately argued and includes good references, it lacks of some important aspect:
Had the Authors verified the efficiency of medium to isolate Campylobacter spp. Using fortified samples or naturally contaminated samples?
Perhaps the Authors suggest to use Campyair only for the growth of isolated strains, in this assumption I suggest to Authors to include this aspect in the discussion
Author Response
The Authors proposed interesting research on the use of a new medium for Campylobacter spp. Isolation without the need for microaerobic atmosphere. Despite the paper is adequately argued and includes good references, it lacks of some important aspect:
Answer: Thank you for your positive comments on our work. The changes suggested have been considered and the modifications don have been highlighted in green.
Had the Authors verified the efficiency of medium to isolate Campylobacter spp. Using fortified samples or naturally contaminated samples?
Perhaps the Authors suggest to use Campyair only for the growth of isolated strains, in this assumption I suggest to Authors to include this aspect in the discussion
Answer: Yes, a prepared faecal sample, inoculated with C. jejuni was used to inoculate CAMPYAIR. Campylobacter grew on the medium, but the growth of other bacteria was inhibited (naturally or artificially present in the faecal sample) was observed. This information was included now in materials and methods, results, and discussion of the new version.
In addition, it was previously discussed that the capacity of CAMPYAIR to recover Campylobacter spp. from complex matrices such as human faces or foods should be assessed in future studies in comparison with other selective medium commonly used. We completed this, indicating that the capacity of CAMPYAIR to recover field isolates Campylobacter spp. from complex matrices should be assessed by future studies.
Reviewer 3 Report
The topic raised by the Authors is very important.
Growing Campylobacter spp. bacteria under aerobic conditions will greatly facilitate its detection. Because in many laboratories these bacteria are not detected due to breeding problems. And it is known that Campylobacter spp. is responsible for the vast majority of diarrhea. Therefore, it is important to detect it because of the complications it can cause.
Introduction, the Authors should extend the information on the risk of infections caused by Campylobacter spp. Apart from complications, they should mention the antibiotic resistance profile of these bacteria.
Moreover, the work contains a few editing errors, eg. line 120-upper index is missing; in Table 3 C.coli "coli" is written with a capital letter. In addition, the descriptions of the tables blend in with the text, which makes the manuscript difficult to read.
Best regards
Author Response
The topic raised by the Authors is very important.
Growing Campylobacter under aerobic conditions will greatly facilitate its detection. Because in many laboratories these bacteria are not detected due to breeding problems. And it is known that Campylobacter spp. is responsible for the vast majority of diarrhea. Therefore, it is important to detect it because of the complications it can cause.
Introduction, the Authors should extend the information on the risk of infections caused by Campylobacter spp. Apart from complications, they should mention the antibiotic resistance profile of these bacteria.
Moreover, the work contains a few editing errors, eg. line 120-upper index is missing; in Table 3 C.coli "coli" is written with a capital letter. In addition, the descriptions of the tables blend in with the text, which makes the manuscript difficult to read.
Answer: Thank you for considering very important our study. Your suggested changes have been highlighted in pink. The description of the tables have been differentiated from the text by reducing its size.
Reviewer 4 Report
The author's describe a newly optimal medium for the isolation of Campylobacter spp. without the need of an microaerobic ambient. The paper is well-written, interesting and reproducible. However I have only some minor issues:
- please include the catalog no. of each product used in your study, since this information helps other scientist to reproduce the data;
- please explain if this medium may be specific for any strains of Campylobacter (jejuni, coli, lari, upsaliensis, etc).
Author Response
The author's describe a newly optimal medium for the isolation of Campylobacter spp. without the need of an microaerobic ambient. The paper is well-written, interesting and reproducible. However I have only some minor issues:
- please include the catalog no. of each product used in your study, since this information helps other scientist to reproduce the data.
- please explain if this medium may be specific for any strains of Campylobacter (jejuni, coli, lari, upsaliensis, etc).
Answer: Thank you for your positive opinion on our work. The catalog number have been added and a sentence has been added to discussion on the species that can grow on this medium. Actually, we only tested C. jejuni and C. coli. Therefore, the growth of other species is also something that should be assessed in future studies. Your suggested changes have been highlighted in grey.
Round 2
Reviewer 2 Report
The Authors had answered to my suggestion exhaustively, the paper is complete and clear.
Now, it can be accepted in present form